# The Impacts of Supplemental Protein during Development on Amino Acid Concentrations in the Uterus and Pregnancy Outcomes of Angus Heifers

**DOI:** 10.3390/ani13121995

**Published:** 2023-06-15

**Authors:** Kiernan J. Brandt, Taylor B. Ault-Seay, Rebecca R. Payton, Liesel G. Schneider, J. Lannett Edwards, Phillip R. Myer, Justin D. Rhinehart, Kyle J. McLean

**Affiliations:** Department of Animal Science, University of Tennessee, Knoxville, TN 37996, USA

**Keywords:** amino acids, heifer development, reproductive efficiency, supplemental protein, uterine environment

## Abstract

**Simple Summary:**

Replacement heifer development is a critical component in beef production. The identification of the ideal uterine environment will greatly benefit reproductive efficiency. Additionally, the elucidation of how diet can influence this environment could alter management strategies. Therefore, our hypothesis was that different levels of protein supplementation would affect the uterine environment of beef heifers without inhibiting development or the ability to conceive. Supplemental protein did not greatly affect the uterine environment as it pertained to amino acid concentrations. However, uterine amino acid concentrations did change throughout development, and protein supplementation can influence uterine luminal fluid composition on d 14 post-insemination, which may affect conception rates.

**Abstract:**

Replacement heifer development is one of the most critical components in beef production. The composition of the ideal uterine environment could maximize fertility and reproductive efficiency. Our hypothesis was that protein supplementation would affect the uterine environment of beef heifers without inhibiting development or reproduction. To test the effects of dietary supplementation on these outcomes, a randomized complete block design with repeated measures was implemented. Angus heifers (*n* = 60) were blocked by body weight (BW) and randomly assigned to one of three supplemental protein treatment groups (10% (CON), 20% (P20), and 40% (P40)). Mixed model ANOVAs were used to determine whether protein supplementation treatments, time, and the interaction or protein supplementation, semen exposure, and the interaction influenced uterine luminal fluid (ULF) and pregnancy outcomes. Amino acids (AAs) were impacted (*p* < 0.001), specifically, the essential AAs: Arg, Iso, Leu, Val, His, Lys, Met, Phe, Trp. Protein supplementation influenced multiple AAs post-insemination: Arg (*p* = 0.03), CC (*p* = 0.05), 1-MH (*p* = 0.001), and Orn (*p* = 0.03). In conclusion, protein supplementation did not affect the reproductive development via puberty attainment or the timing of conception even with alterations in growth. However, uterine AA concentrations did change throughout development and protein supplementation influenced ULF d 14 post-insemination, which may affect the conception rates.

## 1. Introduction

Replacement heifer development is one of the most critical components of beef production with nearly 6 million entering the United States beef herd annually [1]. The overarching goal of development strategies is to maximize the percentage of heifers that conceive early in the first breeding season and maintain a pregnancy. Replacement heifers should weigh 50–65% of their mature body weight (BW; [2,3]) to maximize the number of cyclic females entering the breeding season since puberty is primarily dictated by BW [4,5,6]. Heifers conceiving earlier in the first breeding season have a longer recovery time before the next breeding season, resulting in improved overall productivity and longevity in the herd [7]. Replacement females must incorporate both genetic progress and phenotypic traits to maximize production. During development, a positive correlation between body condition score (BCS), muscle deposition, and reproductive success has been observed, if heifers do not become over-conditioned [8] or to heavily muscled [9]. Heifers supplemented with additional protein experience elevated growth and fertility rates compared to counterparts developed without protein supplementation [10]. Conversely, overfeeding protein can delay the onset of puberty, reduce pregnancy rates at first service, and cause an abnormally low uterine pH which may negatively impact fertility [11,12,13,14].

In addition to influencing growth and development, supplementation may also influence the uterine environment of replacement heifers [12,15]. By identifying the ideal uterine environment in pre-breeding heifers, maximal productivity via increased fertility and reproductive efficiency can be obtained. Developing management practices that lead to an ideal uterine environment will allow producers to identify and more efficiently raise successful replacement females. The establishment and maintenance of pregnancy are dependent on the proper interaction between the uterine environment and embryo to support development until hemotropic exchange can be initiated [16]. Deficiencies of energy substrates or key constituents within the uterine environment are a major contributor to embryonic mortality [17]. The largest percentage of embryonic mortality occurs prior to day 14 post-insemination [18]. The uterine environment comprises secretions and transported molecules that can be quantified and described as the maternal contribution to uterine luminal fluid (ULF; [16]). Prior to maternal recognition, which occurs around day 16 of gestation in beef cattle, the embryo is free floating within the uterus and entirely dependent on ULF for nutrients [19].

Several recent studies have analyzed amino acid (AA) concentrations within ULF throughout the estrous cycle [20], during pre-attachment [16,21] and early gestation [15]. Amino acids have been demonstrated to increase prior to embryonic attachment to provide adequate supply to the growing embryo [16,21], and were reduced in the ULF of sub-fertile cattle at day 17 of pregnancy [22], indicating a relationship between AA concentrations and the successful establishment of pregnancy. Crouse and colleagues [15] reported that heifers receiving a restricted diet had altered uteroplacental AA concentrations from day 16–50 of pregnancy, a time point of development in which the embryo is still largely dependent on ULF. Despite the potential relationship between maternal nutrition and reproductive efficiency, a few studies have analyzed maternal nutrition’s impact on AA concentrations in ULF. Therefore, our hypothesis was that different levels of supplemental crude protein (CP) would impact the AA concentrations within the ULF of developing heifers without altering reproductive development; specifically, pubertal onset and overall conception rates. The objectives of the current study were to quantify the AA concentrations and pH in ULF while meeting developmental goals and utilizing thresholds that are consistent with the industry standards.

## 2. Materials and Methods

### 2.1. Animal Management and Experimental Design

Commercial Angus heifers (*n* = 60), one month after weaning (266.2 ± 12.2 kg), were utilized for the completion of objectives and sample collection. All heifers had ad libitum access to mixed grass hay (Table 1), trace mineral supplement (CO-OP Supreme Cattle Mineral; Tennessee Farmers’ Cooperative; Lavergne, TN, USA), and water. Nutrient analysis for the grass hay was conducted via near-infrared spectroscopy. Heifers received a single experimental supplement for the entirety of the study, provided 4 times per week [23] to maintain an adequate ruminal protein supply. Due to the interaction between energy and protein, an average daily gain of 0.68 kg was targeted to minimize the potential impacts of the different intake levels. Heifers were weighed on two consecutive days, blocked by BW into 4 weight classes, and randomly assigned to one of three supplemental treatments: (1) control (CON); (2) 20% CP supplement (P20); and (3) 40% CP supplement (P40). Supplementation levels averaged 5.19%, 3.80%, and 4.00% of BW for CON, P20, and P40, respectively, and ranged from 2.80% to 7.53% of BW over the course of the treatments. Supplement nutrient composition was conducted via standard analysis wet chemistry (Cumberland Valley Analytical Services; Waynesboro, PA, USA). This resulted in balanced diets calculated to contain a CP of approximately 11, 15, and 19% for CON, P20, and P40 heifers, respectively.

Heifers were housed in three-acre pens with 5 heifers, each resulting in 4 pens per treatment. Bodyweight, BCS (1 = emaciated; 9 = obese; [24]), and a jugular blood samples were collected every two weeks (Figure 1) to monitor the growth and maintain the adequate development of all heifers.

Before the final sample collection, all heifers underwent a 7-day CO-Synch + CIDR estrus synchronization protocol (Figure 1; [25]). Heifers received an initial 100 µg intramuscular (im) injection of gonadotrophin-releasing hormone (GnRH; gonadorelin diacetate tetrahydrate, Cystorelin^®^, Boehrenger Ingleheim, Duluth, GA) and a controlled internal drug release (CIDR) device (1.38 g progesterone; Zoetis Animal Health, Florham Park, NJ, USA) was inserted into the vagina for 7 days. Upon CIDR removal, heifers received an im injection of 500 µg prostaglandin-F2α (PGF2α; Cloprostenol sodium; Synchsure™, Boehrenger Ingleheim, Duluth, GA, USA). Fixed-time artificial insemination was performed by a single technician beginning approximately 54 h after CIDR removal, concurrent with an im injection of GnRH. At the time of breeding, one animal from each pen was randomly selected to not be inseminated and serve as a synchronized but unexposed control for all pregnancy-associated analyses. Immediately following the last uterine flush, all animals were given another injection of PGF2α im to terminate any conceptions that may have survived the 14 d post-insemination flush. Seven days after the end of treatments, all animals underwent another synchronization protocol, AI breeding, and were exposed for 100 d to a fertile bull as per the standard operating procedure of the research unit. Ultrasonography was conducted 35 days after AI and 35 days after bull removal for pregnancy determination. The resulting pregnancies were utilized for the long-term impacts of protein supplementation during development.

### 2.2. Uterine Fluid Collection

Uterine luminal fluid was collected every 28 d from each animal, via a 14 Fr Foley catheter, with the exception of the final sample which was taken 23 d after the 6th uterine flush and 14 d post-insemination resulting in a total of 7 samples/animal (Figure 1). The first ULF sample was collected at the initiation of supplementation. Samples were obtained by passing the catheter through the cervix and injecting 20 mL of sterile saline (Baxter Healthcare Corporation, Deerfield, IL, USA) into the uterine lumen. Saline was mixed throughout the uterus for approximately 45 s before being withdrawn into a syringe using the massage/palpation of the uterus. The saline pH for each bag was recorded prior to using for flushes for comparison back to recovered fluid (Saline + ULF). The pH of ULF was immediately recorded upon removal. The initial pH of saline used for uterine flushes was subtracted from the final pH after flushing to normalize values and was reported as pH change. This calculation reflects the overall change in saline pH after exposure to the uterine environment and mixing with ULF. Following pH measurements, uterine fluid samples were vortexed for 10 s, separated into 2 mL aliquots, and stored at −80℃ until analysis.

### 2.3. Progesterone and Puberty

Plasma samples were collected via jugular venipuncture into 10 mL vacutainer tubes (Becton, Dickinson and Company, Franklin Lakes, NJ, USA) and allowed to clot at room temperature before being centrifuged at 5000× *g* for 20 min at 4 °C. Plasma was separated from blood constituents and stored in 2 mL aliquots at −80 °C. The plasma samples were then analyzed to quantify the progesterone (P4) concentration via radioimmunoassay using the double-antibody radioimmunoassay kit (MP Biomedicals, Irvine, CA, USA) as described previously [26] to estimate the onset of puberty. The onset of puberty was determined when P4 exceeded 1.0 ng/mL in two consecutive samples that also coincided with a normal estrous cycle ranges. Birthdates for each animal were used in conjunction with P4 data to determine the age at puberty. The intra-assay coefficient of variation was 5.22% and the assay sensitivity was 0.08 ng/mL. Seven heifers were removed from puberty analyses due to the discrepancies involving their P4 levels and an inability to establish a cyclical pattern and, thus, determine the onset of puberty.

### 2.4. Amino Acid Analysis

Twenty-five AAs and intermediary metabolites, namely alanine (Ala); arginine (Arg); aspartic acid (Asp); citrulline (Cit); cysteine (CC); glutamine (Gln); glycine (Gly); histidine (His); isoleucine (Ile); leucine (Leu); lysine (Lys); methionine (Met); phenylalanine (Phe); proline (Pro); serine (Ser); threonine (Thr); tryptophan (Trp); tyrosine (Tyr); valine (Val) and intermediates 1-methyl-histidine (1 MH); alpha-aminoadipic acid (AAAA); alpha-aminobutyric acid (AABA); delta-hydroxylysine (DHL); ornithine (Orn); and sarcosine (Sar), were quantified within ULF via liquid chromatography–mass spectrometry (LC/MS). Prior to LC/MS, solid phase extraction was performed by the EZfaast Amino Acid Analysis Kit (Phenomenex, Torrance, CA, USA). Following solid phase extraction, AAs were derivatized in solution for further separation from components that may interfere with quantification. After derivatization, samples were re-dissolved in an aqueous mobile phase before analysis and quantification on an LC/MS system. The column used for LC/MS was the EZ:faast AAA-MS column 250 × 2.0 mm with a flow rate of 0.25 mL/min at 35 ℃. Concentrations were determined by comparing the area under the curve for each peak to certified standards, included with the extraction kit (EZ:faast; sensitivity 1 nmol/mL).

### 2.5. Statistical Analyses

To test the effects of dietary supplementation on heifer BW, BCS, the incidence of puberty, ULF pH, and ULF AA profile, a randomized complete block design with repeated measures was utilized. Because BW, BCS, and ULF were collected from each animal, an individual heifer represents the sampling unit. However, the experimental unit was the pen (*n* = 12; 4 per treatment). Separate mixed-model ANOVAs and mean separation via PROC GLIMMIX (SAS 9.4, Cary, NC, USA) were used to determine whether the protein supplementation, time, or the interaction influenced BW, BCS, pH, age at pubertal onset, or AA concentrations. Separate mixed-model ANOVAs and mean separation via PROC GLIMMIX were used to determine whether protein supplementation, exposure to semen, or the interactions influenced AA concentrations after undergoing synchronization and insemination. Crude protein level and sample day were included in the model as fixed effects. Weight class, the interaction of weight class × crude protein level, and the 3-way interaction of weight class × crude protein level × sample day were included in the model as random effects. Bodyweight, BCS, pH, and P4 levels were included in all AA models as covariates and removed if not significant (*p* > 0.05). Log transformation was performed when AA concentrations did not yield normally distributed residuals. Shapiro–Wilk’s statistic (W > 0.8), the Kolmogorov–Smirnov test (K > 0.2), and the visual assessment of residual plots were assessed to assess normality. Reported concentrations and standard error of the mean were back calculated and reported in µM. A correlation analysis, for any AAs found to be impacted by protein supplementation or from exposure to semen, was utilized to assess the connections between concentrations during development and after insemination. The frequency procedure within SAS was used to determine whether protein supplementation influenced the breeding season following the end of treatments. Due to the limitations of frequency analyses, a survival analysis was also performed to determine whether the time to conception was impacted by previous protein supplementation. Means were considered different when *p* ≤ 0.05 with a tendency at *p* ≤ 0.10.

## 3. Results

### 3.1. Body Weight and Body Condition Score

There was a significant treatment × day interaction (*p* < 0.001) for BW during development. The BW of CON heifers was lower (291.1 ± 12.7 kg; *p* = 0.02) than P20 or P40 heifers (306.7 ± 12.7 kg and 314.6 ± 12.7 kg, respectively) beginning on day 42 and continued through day 163 (338.4 ± 27.82 kg, 376.5 ± 27.82 kg, 377.9 ± 27.82 kg for CON, P20 and P40 heifers, respectively, as shown in Figure 2). Body energy reserves (i.e., BCS) were impacted by the interaction of day × treatment (*p* < 0.001; Figure 3). Average BCS for CON, P20, and P40 heifers was 5.4 ± 0.1, 5.5 ± 0.1, and 5.6 ± 0.1, respectively, but fluctuated more over time than BW. The initial BCS for all groups was between 5.1 and 5.2. Heifers on the P20 and P40 treatments had higher BCS (5.3 ± 0.1 and 5.4 ± 0.1, respectively) than CON heifers (5.0 ± 0.1) on d 28 (*p* < 0.001) but were similar on d 42 (*p* = 0.63). On d 56, P20 supplemented heifers were intermediary (*p* = 0.05) between CON and P40 groups (CON = 5.3 ± 0.1, P20 = 5.5 ± 0.1, P40 = 5.5 ± 0.1) which tended to continue through d 70 (CON = 5.4 ± 0.1, P20 = 5.5 ± 0.1, P40 = 5.6 ± 0.1; *p* = 0.09). Heifers provided P20 and P40 had a greater BCS (5.8 ± 0.1) than CON heifers (5.5 ± 0.1) on d 84 and maintained this difference through d 98 (*p* = 0.002). Supplementation with P20 resulted in higher BCS (5.7 ± 0.1) compared with CON or P40 groups (5.3 ± 0.1 and 5.5 ± 0.1, respectively) on d 112 (*p* < 0.001). Heifers in the P20 and P40 had a higher BCS (5.7 ± 0.1 and 5.9 ± 0.1, respectively) versus CON heifers (5.4 ± 0.1) on d 126 (*p* < 0.001). Heifers supplemented with the P40 supplement had a greater (*p* < 0.001) BCS (5.9 ± 0.1) compared with the CON or P20 groups (5.5 ± 0.1; 5.6 ± 0.1, respectively) on d 140. On the final day of treatment, BCS for P20 and P40 heifers (5.8 ± 0.1 and 5.8 ± 0.1, respectively) was greater (*p* < 0.001) compared with heifers on the CON supplement (5.5 ± 0.1 Figure 3). The impacts of protein supplementation on BW and BCS suggest that growth was limited by the lower protein content in the CON supplement.

### 3.2. Age and Plasma Progesterone at Puberty

Although heifers receiving the CON supplement weighed less than contemporaries throughout most of the study, treatment did not influence age at puberty (*p* > 0.98). The onset of puberty occurred at 340.9 ± 16.3, 344.3 ± 15.0, and 344.0 ± 14.4 d of age for CON, P20, and P40 heifers, respectively, after approximately 100 days of supplementation. While still above the threshold for a functional corpus luteum (1 ng/mL) for all treatments, plasma P4 levels at the time of estimated puberty (Figure 4) were greater (*p* = 0.02) in P20 heifers compared with P40 heifers and CON heifers being intermediate. These data indicate that, while growth may have been impacted by protein supplementation, reproductive maturation and thus heifer development were not impacted.

### 3.3. Uterine pH

Due to the variability of pH amongst the 34—1 L bags of saline, pH was normalized based on flush saline pH. Overall, the pH (6.61 ± 0.2) among samples did not differ (*p* = 0.23). There was an interaction between treatment and time for the relative change in uterine pH (*p* = 0.002; Figure 5). Calculating a relative change in pH accounted for the confounding effect of differences in initial saline pH while still observing pH changes in the uterine environment. On d 28, the P40 group exhibited less relative change in pH compared to the CON group (0.33 ± 0.1 vs. 0.74 ± 0.1; *p* = 0.01), whereas on d 56, the P40 heifers (0.23 ± 0.1) had a more basic uterine environment compared to the P20 (−0.22 ± 0.1) and CON (−0.13 ± 0.1) groups (*p* = 0.005). Over the final four sampling periods (d 84–d 163), all heifers exhibited a basic relative pH but no differences were observed amongst treatments (*p* > 0.10). Relative uterine pH at the time of detectable puberty was also not different among treatments (*p* = 0.18). The lack of impacts on uterine pH could be a consequence of differing estrus status as these females reached puberty or that overall protein levels in the diet were not different enough to have an impact on the reproductive tract.

### 3.4. Amino Acids during Development

Among the free AAs and intermediates quantified from the EZ:faast Amino Acid Analysis Kit, most were not influenced by protein supplementation and none exhibited a treatment × day interaction (*p* ≥ 0.16). However, Cit and DHL tended to differ (*p* = 0.08 and 0.10, respectively) depending on the CP supplementation level. Heifers on the CON supplement tended (*p* = 0.08) to have less Cit (0.07 ± 0.03 µM) compared with heifers given the P40 supplement (0.14 ± 0.03 µM) with P20 heifers being intermediate (0.09 ± 0.03 µM). Uterine DHL tended (*p* = 0.10) to be greater in the ULF of P40 heifers (0.08 ± 0.01 µM) compared with P20 and CON heifers (0.06 ± 0.01 and 0.06 ± 0.01 µM, respectively).

The free AAs that were negatively influenced by BCS and positively influenced by both pH and P4 (as covariates) are presented in Table 2. Arginine concentrations peaked (*p* < 0.001) on d 56 and decreased slightly thereafter. Citrulline was present in low levels throughout development (Cit < 0.19 ± 0.04 µM) but increased (*p* < 0.001) after 28 d (0.11 ± 0.04 µM) of supplementation, except on d 112 (0.01± 0.04 µM), which was similar to initial concentrations. Concentrations of Gln decreased (*p* < 0.001) from d 28 to 112 (1.50 ± 0.22 µM to 0.05 ± 0.22 µM) but were highest at d 140 (1.32 ± 0.22 µM). Free Orn concentrations varied over time (*p* < 0.001), but only exceeded 0.1 µM on d 28. Proline ranged from 0.04 to 0.69 ± 0.09 µM (*p* < 0.001; Table 2).

Body condition score negatively influenced but pH positively impacted AA concentrations in Table 3. Concentrations of Aaba were very low, never exceeding 1 µM, but did fluctuate throughout development (*p* < 0.001). Glycine concentrations were increased (*p* < 0.001) on d 84 and 140 (4.46 ± 3.56 µM and 11.47 ± 3.56 µM, respectively, as shown in Table 3). The isoleucine peaked (*p* < 0.001) early (d 28 and 84) and decreased during the remainder of the development. Leucine was the highest (*p* < 0.001) at d 28 (0.62 ± 0.09 µM) and remained elevated for the remainder of development (Table 3). Concentrations of 1 MH were present in very low quantities (1 MH < 0.08 ± 0.01 µM) but did fluctuate over time (*p* < 0.001) during development. Valine concentrations were greatest (*p* < 0.001) on d 28 (1.35 ± 0.17 µM) but relatively stable (0.76–1.24 µM) except for on d 112 (0.23 ± 0.17 µM; Table 3).

Several AAs were positively influenced by pH as a covariate (Table 4) with the exception of Asp, which was negatively impacted. The concentration of Ala increased after 28 d of supplementation and fluctuated during development (*p* < 0.001; Table 4). Aspartic acid fluctuated throughout the study (*p* < 0.001) but peaked at d 84 (110.91 ± 21.4 µM). Free DHL was very low throughout development but differed across time (*p* < 0.001). Histidine concentrations oscillated (*p* < 0.001) during development with d 28, 84, and 140 being elevated compared to d 0, 56, and 112 (Table 4). Lysine values increased (*p* < 0.001) on d 28 of supplementation, remained elevated through d 56, and decreased thereafter (Table 4).

The changes in AA concentrations during development are intriguing and may potentially be useful to predict fertility or as biomarkers to development but further research needs to be conducted to elucidate these impacts.

### 3.5. Amino Acids after Insemination

On d 163 or 14 post-insemination, no AAs were influenced by the treatment × exposure interaction (*p* > 0.21; Table 5). The exposure to semen, via artificial insemination, did not impact any of the AA concentrations but tended to impact both CC (*p* = 0.07) and Gln (*p* = 0.10; Table 5). Cysteine tended to increase (*p* = 0.07) in heifers that were not exposed to semen (0.41 ± 0.3 µM) compared with heifers that were inseminated (0.03 ± 0.3 µM). Interestingly, Gln tended to increase (*p* = 0.10) in heifers that were inseminated (3.22 ± 1.0 µM) compared with heifers not exposed to semen during breeding (1.30 ± 1.0 µM). Protein supplementation did impact multiple AA post-insemination. Arginine was dramatically increased (*p* = 0.03) in the P40 heifers (18.75 ± 7.74 µM) over CON supplemented heifers (1.88 ± 7.74 µM) and P20 heifers (5.82 ± 7.74 µM) were intermediate (Table 5). Concentrations of CC were increased (*p* = 0.05) in P40 compared with CON and P20. Heifers supplemented with P40 supplementation (0.27 ± 0.07 µM) had increased (*p* = 0.001) 1 MH over CON and P20 (0.11 ± 0.07 and 0.08 ± 0.07 µM, respectively, as shown in Table 5). Ornithine concentrations increased (*p* = 0.03) in P40 heifers (0.43 ± 0.18 µM) compared with CON heifers (0.16 ± 0.18 µM) and P20 were intermediate (0.13 ± 0.18 µM). Lysine tended to be increased (*p* = 0.10) in P40 supplemented heifers (2.89 ± 1.41 µM) over CON and P20 heifers (0.94 ± 1.41 and 1.05 ± 1.41 µM, respectively, as shown in Table 5). The impacts of protein supplementation and exposure to semen on AA concentrations have identified AAs that may be vital to the establishment of pregnancy. These AAs and the pathways in which they function need to be further investigated to establish the entirety of their impacts on pregnancy.

### 3.6. Correlation between Development and Pregnancy

All AAs that were found to be different or tended to differ based on protein supplementation or semen exposure (Arg, Cit, CC, DHL, Gln, Lys, 1 MH, Orn) were utilized in a correlation analysis to determine whether uterine concentrations of AAs during development (d 0, 28, 56, 84, 112, 140, overall average) were indicative of AAs concentrations in ULF post-insemination (d 163; Table 6). Concentrations of DHL, Lys, or Orn on d 163 were not correlated (*p* ≥ 0.12) with concentrations on any d during development or the average concentration. On d 0, Arg tended to be negatively correlated (R = −0.39) with concentrations on d 163 for heifers receiving the P20 supplement. Arginine concentrations on d 56 were positively correlated with d 163 Arg concentration for P20 (R = 0.50; *p* = 0.03) and P40 (R = 0.53; *p* = 0.02) but only tended to be correlated in CON heifers (R = 0.42; *p* = 0.07; Table 6).

Arginine concentrations on d 28, 84, 112, 140, and the overall average were not correlated (*p* ≥ 0.22) with Arg concentrations after breeding. Citrulline concentrations on d 163 were not related (*p* ≥ 0.41) to any Cit concentrations during development for heifers on the P40 treatment. For heifers on the P20 treatment, concentrations of Cit on d 56 were positively correlated (R = 0.68; *p* < 0.01; Table 6) with concentrations post-insemination but were not found to be related on any other day (*p* ≥ 0.15). On d 56 and the overall average concentrations of Cit were positively correlated with Cit concentrations on d 163 (Table 6). Cysteine concentrations post-insemination (d 163) were positively correlated (*p* ≤ 0.05) on d 28, 56, 112, 140, and the overall average and negatively correlated on d 84 (R = −0.56; *p* = 0.01; Table 6) for heifers receiving the P40 supplement. The overall average (R = 0.75) and d 140 (R = 0.60) CC concentrations were positively correlated (*p* < 0.01) with concentrations after breeding but were not related (*p* ≥ 0.21) on d 0, 28, 56, 84, or 112 in P20 heifers. For heifers receiving the CON supplement, CC concentrations after insemination were positively correlated (*p* < 0.05) with CC concentrations on d 28 (R = 0.45) and d 112 (R = 0.59) and tended to be correlated (*p* < 0.10) with concentrations on d 56 (R = 0.37) and the overall average (R = 0.41; Table 6). Glutamine concentration following breeding tended (*p* = 0.08) to be negatively correlated (R = −0.40) with concentrations on d 112 but was not related to concentrations at any other time point or the average (*p* ≥ 0.17). Concentrations of 1 MH after insemination were highly correlated to 1 MH concentrations on d 56 (R = 0.61; *p* < 0.01), d 84 (R = 0.64; *p* < 0.01) and the overall average (R = 0.93; *p* < 0.01), but only in heifers on the CON supplement. Heifers on the P20 and P40 treatments were not found (*p* ≥ 0.31) to have any relationships in 1 MH concentrations during development and after breeding (Table 5). These correlations may indicate that the concentrations of specific AAs during development may be useful as biomarkers to indicate the fertility and development of replacement heifers.

### 3.7. Consequential Impacts of Protein on Pregnancy

To elucidate whether the protein supplementation level had any long-term effects on reproductive function frequency and survival analyses were performed on pregnancy results following treatments (Table 7). There was not a difference (*p* = 0.13) in the frequency distribution of when conception occurred amongst treatments. Interestingly, double the number of CON heifers were conceived under AI, and all were bred by the second cycle, whereas both the P20 and P40 conceptions occurred relatively linearly (Table 7).

The survival analysis did confirm what the frequency analysis did not have the power to determine. The average d to conception was shorter (*p* = 0.02) for heifers on the CON supplement (24.0 ± 4.5 d) compared to P20 heifers (46.1 ± 7.1 d) with P40 being intermediate (39.5 ± 5.8 d; Table 7). These data indicate that AA concentrations and reproductive traits were not highly impacted by protein supplementation; however, there may still be overall influences that could alter the reproductive efficiency of replacement heifers.

## 4. Discussion

In the current experiment, heifers receiving CON supplement achieved puberty at approximately the same age as contemporaries, despite having a lower BW. This is in agreement with a previous study where heifers receiving a high starch intake had a lower BW at puberty [27]. Age at which puberty occurs in developing replacement heifers in relation to the beginning of the breeding season has a large impact on the lifetime productivity of these animals. Day and Nogueira [7] determined that heifers which conceived early in their first breeding season were more productive and economically efficient than females conceiving later in their first breeding season. This concept reinforced the importance of developing females that are pubertal and cycling before the breeding season [28]. Previous research has demonstrated that heifers bred after at least one previous ovulation experience elevated conception rates compared with heifers bred at first estrus [29]. Traditionally, producers have developed heifers targeting a BW threshold of 60–65% of mature BW [30] in the hopes of maximizing the number of cycling females at the beginning of the breeding season. Protein was demonstrated as a limiting factor for growth during development in CON supplemented groups, despite adequate energy for a maintained positive plane of nutrition in the current study. This is likely responsible for the decreased BW attained by CON supplemented groups during the latter portion of the trial, despite intake adjustment attempts to maintain similar rates of gain between treatments. When developing replacement heifers, BCS at the beginning of the breeding season is critical, as reductions in fertility have been noted for females with a BCS < 5 [31] or > 6 [8]. This BCS goal was attained by all heifers in the current experiment. This likely explains why our model did not influence development in terms of cyclicity, as BCS was highly variable between treatments, but a positive plane of nutrition was maintained for all groups. Heifers receiving the P20 supplement, containing 75% DDGS, had better progesterone than CON or P40 heifers at the detection of puberty, similarly to the results published by Gunn et al., [32], in which feeding DDGS increased the circulating progesterone levels in addition to altering follicular growth and estradiol production. The increased RUP values in addition to the elevated fat content in DDGS were both hypothesized as the mechanism for these changes; however, further research is necessary to improve supplementation strategies [32,33].

In order for proper embryo development to take place, pH within the uterine environment must be closely regulated to maintain developmental competence [34]. A majority of cellular processes are regulated by pH, and dysregulation can inhibit proper development if pH exceeds [34], or falls below [35,36], a physiological intracellular pH of 7.4. During the luteal phase of the estrous cycle, pH of ULF is relatively stable around 7.0–7.2; however, it undergoes an acidic change to approximately 6.8 [12,37,38] during the follicular phase, with the lowest pH measurements recorded near the time of estrus. This change in pH is unique to the uterus during the follicular phase and does not manifest in any other bodily fluid measured in previous studies [12,38]. This decrease in pH is associated with increases in estradiol [37,38] and was speculated to be a potential mechanism for sperm transport [37,38]. The uterine pH returns to approximately 6.8 at estrus, to provide an isotonic environment to spermatozoa [12]. Interestingly enough, the uterine pH was not impacted during the heifer development of the current study. This may be because protein treatments were only supplemental but could also be due to the different estrous cycle stages of these animals. Previous studies [12,37,38] have reported a difference between the follicular and luteal phases of the estrous cycle. Since the heifers in the current study were pre-pubertal at the onset of treatments, the stage of the estrous cycle could not be controlled as animal reached puberty naturally and at different times.

Classical studies originally reported an impact of dietary protein on uterine pH, in a study that supplied cycling dairy heifers with 150% of crude protein requirements [12]. High protein intake has also been demonstrated to reduce fertility in gilts [39] and ewes [40], as well as cattle [12,41]. Specifically, in cattle, high levels of intake protein resulted in lower uterine pH and subsequent conception rates than controls [12]. Heifers receiving the highest amounts of supplemental protein (P40) in the current experiment did have a more acidic uterine environment than other treatments in the second sample. This was the only instance where higher levels of protein supplementation coincided with reduced uterine pH and may indicate an adaptive mechanism to long-term protein supplementation. The results of the current study are similar to those observed by Grant et al. [42] where high levels of N supplementation failed to elicit a decrease in uterine pH during the luteal phase, and actually caused a subtle increase in uterine pH. This reinforces the concept that heifers receiving higher levels of supplemental protein had adequate amounts for maintained growth and development, without the negative impacts on fertility historically associated with higher amounts of protein supplementation.

Overall, AA concentrations in the present study were much lower than those reported in previous studies using similar flush methods to analyze ULF [15,16,20,21,43]. Unlike previous studies, the kit utilized for AA extraction measured only free AAs and intermediary metabolites, while allowing larger proteins and other interfering compounds to pass through in the wash solution. In previous studies, heifers were all cyclic prior to AA analysis. In the current study, nearly 40% of heifers did not achieve cyclicity until the final sample, 14 days post-insemination, following estrus synchronization. These results indicate that AAs may increase in prevalence after puberty and their transporters may not become active within the reproductive tract until cyclicity has been established. Of the 25 examined AAs and their intermediates, 18 were at their highest at sample 7. Previous studies in cycling heifers have shown Gly to be the most prevalent AAs in ULF [15,16,20]. In the current study, Gly was found in the highest concentration at sample 7 when most heifers achieved cyclicity. The AAs Ala, Arg, CC, Gln, Gly, Ser, Val, and the intermediary metabolite AAAA were found in the highest concentrations in the current study, in agreement with previous studies [15,20,21], suggesting these AAs may be utilized most readily by the embryo upon arrival in the uterine environment. Amino acids were consistently detected across 6 of the 7 samples. Sample 5 had the lowest concentration for 23 of the 25 examined AAs, with the exception of AAAA and CC, although further investigation is necessary to elucidate the cause of these low concentrations. Protein supplementation affected the concentrations of two AAs, one of which had the highest values for P40 supplemented heifers (Trp). These observations are in agreement with a study [44] which found that heifers supplemented with different protein sources had different levels of Phe and Leu in serum. Given that AAs in ULF are higher than they are in blood [20], it is logical that AA differences were detected in the current study. The combination of different feedstuffs and increasing protein levels in the supplements may be the cause of the differences in our dataset.

The composition of the histotroph, specifically in terms of AAs, is important in fetal development [45,46]. These AAs provide key components that can be used as fuel for growth and development [47], osmoregulation [48], and cell proliferation [49]. Maternal nutrition during early gestation will alter the AAs in the histotroph [15]. The current dataset identified some of the same AAs (Arg, Cit, CC, Gln, Lys, and Orn) that could be impacted by limited protein differences. Neutral AAs, in the uterus, have been reported in greater concentrations earlier in gestation [16], but then the uterine environment transitions to others, specifically Arg and Gln, as gestation progresses in cattle [15] and sheep [50]. The identification of these AAs impacted by limited protein differences confirms the importance of these AAs and the correlations to concentrations during replacement heifer development found in the current study may implicate these AAs as biomarkers for reproductive efficiency. However, more direct research needs to be conducted to elucidate the efficacy of this potential.

The importance of early conception in the first breeding season largely affects the lifetime production of replacement heifers. Day and Nogueira [7] determined that heifers conceiving early in the breeding season weaned heavier calves and were more likely to become pregnant as 2-year-old cows. Thus, meeting one of the primary reproductive targets for a beef cowherd, namely ensuring that heifers calve prior to 2 years of age, serves to optimize long-term productivity [51]. These benchmarks for long-term success indicate the need for high fertility among replacement heifers at the start of the breeding season, and the importance of promoting sexual maturity at a younger age. In the current study, protein supplementation did not influence the attainment of puberty. However, CON heifers did conceive earlier in the breeding season by 22 d, which could influence longevity in the beef herd. Several studies have demonstrated an improvement in conception rates in subsequent estrus events compared to that of their first cycle [29,52]. Recent data have reported that heifers exhibiting at least two cycles prior to breeding are more likely to conceive in the first 21 days of the breeding season [52], further emphasizing the importance of reducing the age at puberty. Additionally, heifers breed at their third estrus attained 21% higher conception rates than those bred at the first estrus [29]. While the decrease in time to conception could be an artifact of estrus manipulations, this could also indicate the impact of protein supplementation during heifer development.

## 5. Conclusions

In summary, protein supplementation influenced the BW and BCS but did not affect age at puberty. Variations in the final BW or BCS were not intended; however, these results are useful in signifying the importance of the overall nutritional plane since CON heifers were gaining weight and maintained an adequate BCS for successful heifer development. Protein levels in the CON diet do appear to be limiting growth. However, it is likely that increased supplementation to CON heifers to compensate for decreased growth would elevate the overall feed costs without an increased return on production efficiency. Further research is necessary to determine whether a lower BW at puberty, achieved by a lower ADG, can be achieved using supplements similar to P20 or P40 without compromising the reproductive success but decreasing the cost of development. The supplement type did affect the amount of P4 at the attainment of puberty and altered the concentrations of Trp and Orn in ULF of developing heifers. Thus, this supports the hypothesis that supplements with different characteristics, in this case, CP, are transported post-digestion in varying amounts into the uterine environment. Further studies are needed to determine whether AA concentrations are consistently affected by different feed types and stages of development. Additionally, identifying the times of development and AAs that may affect fertility could be beneficial to reproductive and production efficiencies in beef cattle. These effects could be elucidated to develop management strategies to optimize heifer development by decreasing the cost associated with development while maximizing reproductive and productive outputs.

## Figures and Tables

**Figure 1 animals-13-01995-f001:**
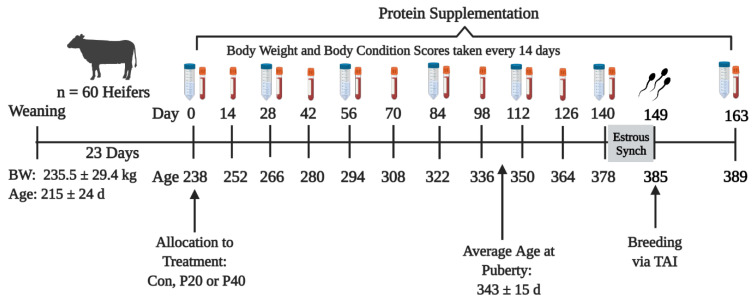
Project timeline for protein supplementation (initiation of supplementation = day 0, blood samples for puberty analyses, and uterine flushes for amino acid quantification. Age is represented as the overall average ± 24 d. Uterine flush collections are represented by the conical tubes and plasma collections represented by the blood tubes.

**Figure 2 animals-13-01995-f002:**
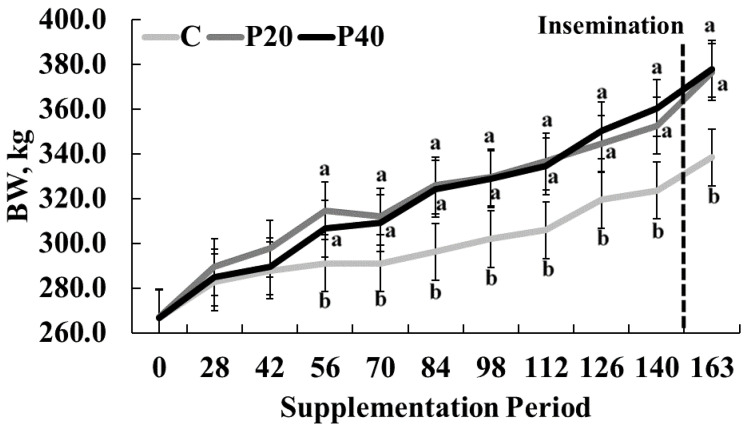
Average bodyweight (BW), in kg, of each treatment (control crude protein supplement (CON), 20% crude protein supplement (P20), 40% crude protein supplement (P40)) at each sample collection day (0–140) over the duration of supplementation. The dashed vertical line indicated the time at which insemination occurred. Means with different letters indicate significant differences (*p* < 0.05) on a given d.

**Figure 3 animals-13-01995-f003:**
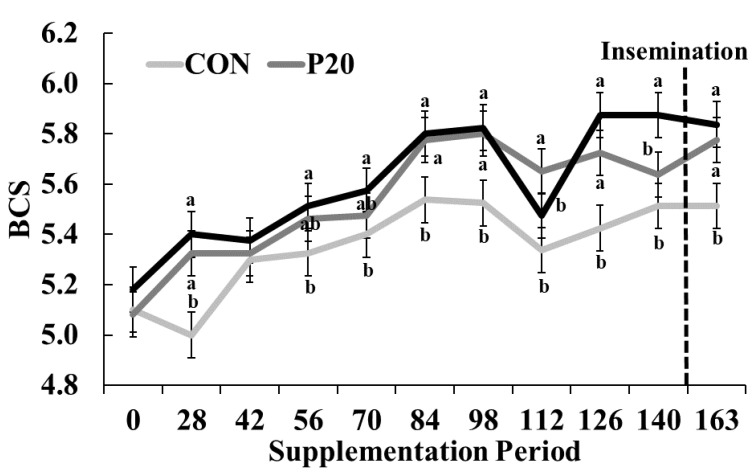
Changes in body condition score (BCS) by treatment (control crude protein supplement (CON), 20% crude protein supplement (P20), and 40% crude protein supplement (P40)) over the duration of supplementation. The dashed vertical line indicated the time at which insemination occurred. Means with different letters indicate the significant differences (*p* < 0.05) on a given d.

**Figure 4 animals-13-01995-f004:**
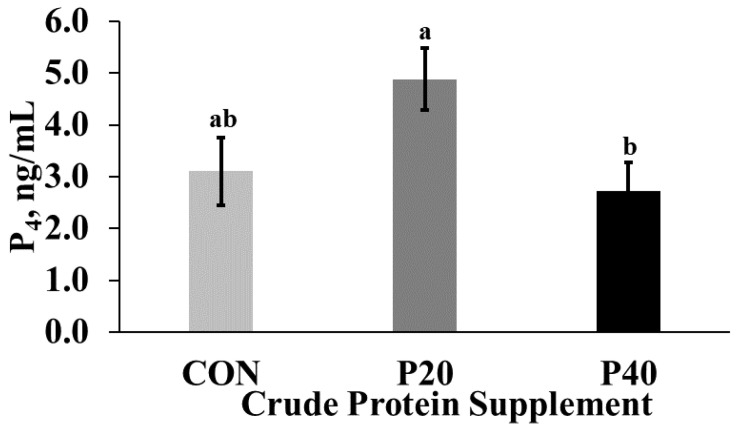
Average progesterone (ng/mL) content in blood plasma at the detection of pubertal onset by treatment ((control crude protein supplement (CON), 20% crude protein supplement (P20), 40% crude protein supplement (P40)). Means with different letters indicate significant differences between treatments (*p* = 0.02).

**Figure 5 animals-13-01995-f005:**
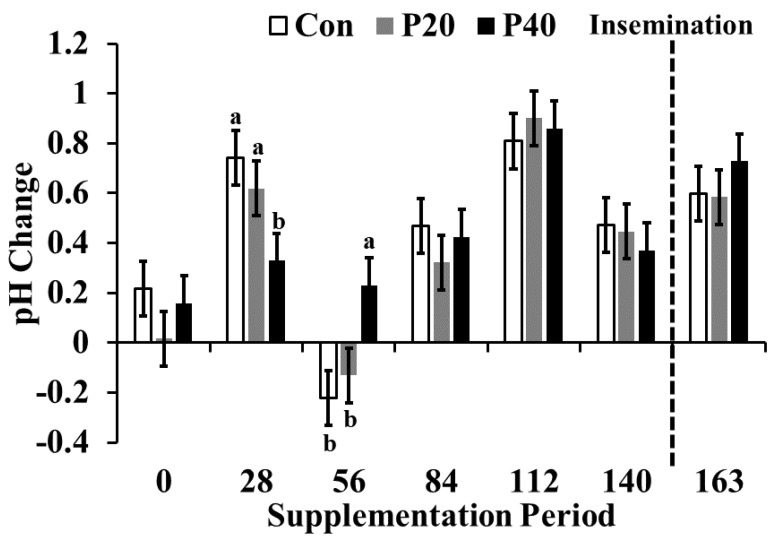
Average change in uterine pH between treatments at each sample collection during the supplementation period for each treatment (control crude protein supplement (CON), 20% crude protein supplement (P20), 40% crude protein supplement (P40)) calculated by subtracting initial pH of flush saline from final uterine flush pH to quantify the uterine environment’s impact on pH, as well to correct for variability in the saline pH. The dashed vertical line indicates the time at which insemination occurred. Means with different letters indicate significant differences between treatments (*p* < 0.05).

**Table 1 animals-13-01995-t001:** Nutrient concentrations of hay and dietary supplements.

Feed	Composition	Dry Matter (%)	Crude Protein (%)	Fat (%)	RUP ^1^ (%)
Hay	100% Grass Hay	95.6	7.6	1.97	---
Con ^2^	100% Cracked Corn	87.5	8.2	3.77	60.8
P20 ^3^	25% Cracked Corn 75% DDG ^4^	91.6	23.6	7.40	75.1
P40 ^5^	25% DDG 75% Soybean Meal	90.2	47.1	2.80	45.9

^1^ RUP: rumen undegraded protein; ^2^ Con: control supplement; ^3^ P20: 20% protein supplement; ^4^ DDG: dried distillers’ grains; ^5^ P40: 40% protein supplement.

**Table 2 animals-13-01995-t002:** Amino acid (AA) concentrations in the uterine fluid of heifers during development ^1,2^.

AAs	Day of Sampling							*p* Value		
	0	28	56	84	112	140	SEM	Trt ^3^	Day	Trt × Day
Arginine (Arg)	0.65 ^cd^	0.95 ^bc^	4.43 ^a^	3.65 ^ab^	0.031 ^d^	2.39 ^ab^	0.60	0.27	<0.001	0.27
Citrulline (Cit)	0.03 ^bc^	0.11 ^a^	0.19 ^a^	0.14 ^a^	0.01 ^c^	0.16 ^ab^	0.04	0.08	<0.001	0.60
Glutamine (Gln)	0.77 ^b^	1.50 ^a^	1.07 ^ab^	0.97 ^ab^	0.05 ^c^	1.32 ^ab^	0.22	0.24	<0.001	0.99
Ornithine (Orn)	0.03 ^bc^	0.14 ^a^	0.09 ^ab^	0.08 ^ab^	0.01 ^c^	0.09 ^ab^	0.03	0.21	<0.001	0.95
Proline (Pro)	0.21 ^cd^	0.47 ^ab^	0.40 ^bc^	0.69 ^a^	0.04 ^d^	0.30 ^bc^	0.09	0.74	<0.001	0.80

^1^ All data analyzed within this table were significantly negatively influenced by BCS, and positively influenced by pH and progesterone. ^2^ All data analyzed within this table were log transformed but are presented as untransformed means and SEM; ^3^ Trt: protein supplementation treatment; ^a,b,c,d^ means without a common superscript differ by *p* < 0.05.

**Table 3 animals-13-01995-t003:** Changes during the development of amino acid (AA) concentrations in the uterine fluid of heifers ^1,2^.

AAs	Day of Sampling							*p* Value		
	0	28	56	84	112	140	SEM	Trt ^3^	Day	Trt × Day
Alpha-Aminobutyric Acid (Aaba) ^4^	0.11 ^c^	0.70 ^a^	0.30 ^b^	0.92 ^a^	0.13 ^c^	0.45 ^b^	0.08	0.66	<0.001	0.93
Glycine (Gly)	1.52 ^b^	1.81 ^ab^	1.94 ^ab^	4.46 ^a^	1.72 ^b^	11.47 ^a^	3.56	0.99	0.001	0.64
Isoleucine (Ile)	0.22 ^cd^	0.67 ^a^	0.37 ^bc^	0.57 ^ab^	0.07 ^d^	0.32 ^c^	0.08	0.72	<0.001	0.88
Leucine (Leu)	0.22 ^bc^	0.64 ^a^	0.43 ^b^	0.46 ^ab^	0.15 ^c^	0.39 ^b^	0.09	0.59	<0.001	0.65
1-Methyl-Histidine (1 MH)	0.01 ^bc^	0.02 ^bc^	0.04 ^b^	0.08 ^a^	0.01 ^c^	0.03 ^bc^	0.01	0.35	<0.001	0.83
Valine (Val)	0.76 ^b^	1.35 ^a^	1.15 ^ab^	1.24 ^a^	0.23 ^c^	1.11 ^ab^	0.17	0.49	<0.001	0.53

^1^ All data analyzed within this table were significantly negatively influenced by BCS, and positively influenced by pH; ^2^ All data analyzed within this table were log transformed but are presented as untransformed means and SEM; ^3^ Trt: protein supplementation treatment; ^4^ Aaba was determined to be normally distributed. Thus, Aaba was analyzed as untransformed data and was never log transformed; ^a,b,c,d^ means without a common superscript differ by *p* < 0.05.

**Table 4 animals-13-01995-t004:** The effects of pH on amino acid (AA) concentrations in the uterine fluid of heifers during development ^1,2^.

AAs	Day of Sampling							*p* Value		
	0	28	56	84	112	140	SEM	Trt ^3^	Day	Trt × Day
Alanine (Ala)	0.58 ^c^	1.93 ^a^	1.29 ^b^	2.02 ^a^	0.23 ^c^	1.40 ^b^	0.22	0.60	<0.001	0.95
Aspartic Acid (Asp) ^4^	9.53 ^b^	3.67 ^b^	1.96 ^b^	110.91 ^a^	11.36 ^b^	26.52 ^a^	21.4	0.13	<0.001	0.56
Delta-hydroxylysine (Dhl)	0.03 ^d^	0.08 ^b^	0.07 ^bc^	0.14 ^a^	0.03 ^cd^	0.05 ^bcd^	0.01	0.10	<0.001	0.95
Histidine (His)	0.09 ^c^	0.73 ^a^	0.15 ^bc^	0.47 ^ab^	0.03 ^c^	1.15 ^ab^	0.24	0.50	<0.001	0.83
Lysine (Lys)	0.21 ^bcd^	0.42 ^a^	0.37 ^ab^	0.19 ^abc^	0.02 ^d^	0.12 ^cd^	0.08	0.64	<0.001	0.95
Methionine (Met)	0.04 ^bc^	0.07 ^ab^	0.08 ^ab^	0.07 ^ab^	0.01 ^c^	0.14 ^a^	0.02	0.76	<0.001	0.64
Phenylalanine (Phe)	0.07 ^bc^	0.18 ^a^	0.17 ^ab^	0.13 ^abc^	0.05 ^c^	0.13 ^abc^	0.03	0.47	<0.001	0.50
Sarcosine (Sar)	0.10 ^b^	0.15 ^ab^	0.18 ^ab^	0.19 ^a^	0.02 ^c^	0.08 ^bc^	0.02	0.90	<0.001	0.58
Tryptophan (Trp)	0.04 ^bc^	0.09 ^a^	0.06 ^ab^	0.07 ^ab^	0.01 ^c^	0.04 ^bc^	0.01	0.22	<0.001	0.51
Tyrosine (Tyr)	0.06 ^bc^	0.13 ^ab^	0.14 ^ab^	0.09 ^ab^	0.01 ^c^	0.28 ^a^	0.05	0.54	<0.001	0.45

^1^ All data analyzed within this table were significantly positively influenced by pH; ^2^ All data analyzed within this table were log transformed but are presented as untransformed means and SEM; ^3^ Trt: protein supplementation treatment; ^4^ Asp was the only AA in this dataset that was negatively influenced by pH; ^a,b,c,d^ means without a common superscript differ by *p* < 0.05.

**Table 5 animals-13-01995-t005:** Amino acid (AA) concentrations in the uterine fluid of heifers based on protein supplementation after breeding ^1^.

AAs	Treatment				Semen Exposure			*p* Value	
	CON	P20	P40	SEM	Yes	No	SEM	Trt ^2^	Exposure ^3^
Alanine (Ala; pH)	0.72	1.82	1.56	0.89	1.34	1.21	0.75	0.42	0.86
Arginine (Arg; pH)	1.88 ^a^	5.82 ^ab^	18.75 ^b^	7.74	6.36	8.77	7.02	0.03	0.69
Citrulline (Cit)	0.10	0.11	0.19	0.09	0.16	0.10	0.08	0.69	0.51
Cysteine (CC; BCS)	0.04	0.26	0.65	0.39	0.03	0.41	0.30	0.05	0.07
Delta-hydroxylysine (Dhl)	0.01	0.05	0.07	0.04	0.04	0.04	0.04	0.45	0.85
Glutamine (Gln)	1.04	3.20	2.58	1.23	3.22	1.30	1.00	0.26	0.10
Glycine (Gly; pH)	6.45	22.63	9.73	9.08	17.56	7.20	7.73	0.24	0.13
Histidine (His; BCS, pH)	0.19	0.27	0.12	0.12	0.18	0.20	0.10	0.61	0.88
Isoleucine (Ile; BCS, pH)	0.26	0.29	0.42	0.21	0.41	0.25	0.17	0.85	0.45
Leucine (Leu; BCS, pH)	0.26	0.50	0.34	0.25	0.39	0.33	0.21	0.79	0.82
Lysine (Lys; pH)	0.94	1.05	2.89	1.41	0.90	1.10	1.19	0.10	0.89
Methionine (Met; pH)	0.05	0.19	0.18	0.08	0.12	0.14	0.07	0.20	0.86
1-Methyl-Histidine (1 MH; BW, BCS, P4)	0.11 ^a^	0.08 ^a^	0.27 ^b^	0.07	0.09	0.07	0.06	0.001	0.87
Ornithine (Orn; pH)	0.16 ^a^	0.13 ^ab^	0.43 ^b^	0.18	0.08	0.18	0.15	0.03	0.57
Phenylalanine (Phe; pH)	0.23	0.59	1.19	0.59	0.41	0.58	0.52	0.16	0.98
Proline (Pro; pH)	0.18	0.67	0.58	0.28	0.40	0.46	0.23	0.23	0.80
Sarcosine (Sar; BCS)	0.14	0.14	0.23	0.11	0.17	0.17	0.10	0.61	0.99
Serine (Ser)	0.35	0.82	0.30	0.44	0.61	0.32	0.35	0.63	0.50
Threonine (Thr)	0.13	0.53	0.32	0.21	0.24	0.36	0.18	0.32	0.58
Tryptophan (Trp; pH)	0.01	0.07	0.12	0.05	0.07	0.05	0.05	0.25	0.64
Tyrosine (Tyr; pH)	0.31	0.58	1.16	0.65	0.45	0.64	0.48	0.19	0.81
Valine (Val; pH)	0.16	0.51	0.62	0.29	0.42	0.35	0.22	0.29	0.81

^1^ All data analyzed within this table were significantly positively influenced by pH and presented as untransformed means and SEM; ^2^ Trt: protein supplementation treatment; ^3^ exposure to semen or not (negative control) at breeding; ^a,b^ means without a common superscript differ by *p* < 0.05.

**Table 6 animals-13-01995-t006:** The impact of protein supplementation on the correlation between uterine amino acid (AA) concentrations during development and uterine AA concentrations following insemination.

AAs ^1^	Trt ^2^	0	28	56	84	112	140	Avg ^3^
Arginine	Con	---	---	0.42 (0.07)	---	---	---	---
	P20	−0.39 (0.09) ^4^	---	0.50 (0.03)	---	---	---	---
	P40	---	---	0.53 (0.02)	---	---	---	---
Citrulline	Con	---	---	0.62 (<0.01)	---	---	---	0.49 (0.03)
	P20	---	---	0.68 (<0.01)	---	---	---	---
Cysteine	Con	---	0.45 (0.05)	0.37 (0.10)	---	0.59 (<0.01)	---	0.41 (0.07)
	P20	---	---	---	---	---	0.75 (<0.01)	0.60 (<0.01)
	P40	---	0.92 (<0.01)	0.51 (0.03)	−0.56 (0.01)	0.45 (0.05)	0.48 (0.04)	0.53 (0.02)
Glutamine	Con	---	---	---	---	---	---	---
	P20	---	---	---	---	−0.40 (0.08)	---	---
	P40	---	---	---	---	---	---	---
1-Methyl-Histidine	Con	---	---	0.61 (<0.01)	0.64 (<0.01)	---	---	0.93 (<0.01)
	P20	---	---	---	---	---	---	---
	P40	---	---	---	---	---	---	---

^1^ All AAs were identified for correlation analyses by a *p* ≤ 0.10 during mean analyses on d 163 following insemination; ^2^ Trt: is the protein supplement level; ^3^ Avg: is the average concentration for each AA over the course of development; ^4^ all values are reported with the R-value outside of the parentheses and the resulting *p*-value inside the parentheses.

**Table 7 animals-13-01995-t007:** Long-term effects of protein supplementation during development on the pregnancy rates in the subsequent breeding season.

Frequency Results	CON	P20	P40	*p*-Value
AI	8	4	4	0.13
1st Cycle	3	4	4	
2nd Cycle	8	4	4	
3rd Cycle	0	3	3	
4th Cycle	0	2	0	
Final Open	0	3	4	
Time to Conception, d ^1^	24.0 ± 4.5	46.1 ± 7.1	39.5 ± 5.8	0.02

^1^ Time to conception analysis was conducting via a survival analyses.

## Data Availability

The data presented in this study are available upon request from the corresponding author.

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
