# Peer review of "The Impacts of Supplemental Protein during Development on Amino Acid Concentrations in the Uterus and Pregnancy Outcomes of Angus Heifers"

_animals, 2023, doi:10.3390/ani13121995_

Round 1

Reviewer 1 Report

This manuscript evaluated the effects of different levels of supplemental crude protein on amino acid concentrations and pH of the uterine luminal fluid, reproductive development, and fertility in beef replacement heifers. Overall, this study and data provide important insights into the influence of protein supplementation on the uterine environment, reproductive development, and pregnancy success in heifers. Evaluation of amino acid concentrations and changes in pH over the development period provide add important data into the changes in the uterine environment that occur during post-weaning development of heifers. The results and discussion are very well written, and the authors did an excellent job explaining the potential implications of their results, as well as areas for additional research. The addition of discussion regarding the changes in pH that occurred early in the supplementation period, as well as differences in post-supplementation conception rates would further improve the overall discussion.

Overall Comments:

Authors should expand discussion of uterine luminal fluid pH changes to include the differences in pH that occurred between days 0-56 of supplementation. Could changes in the pH early in the supplementation period be related to changes in the uterine environment prior to puberty attainment? While the relative pH was similar at detectable puberty, could there be shifts occurring in the period leading up to attainment of puberty?

Authors should provide discussion on the difference in the time to conception post-supplementation. A 22 d difference was noted in the survival analysis for the time to conception between CON and P20 heifers.

Specific comments:

 L17-18: Should define abbreviations for AA and ULF

L21: Sentence “Identifying the ideal uterine environment, fertility and reproductive efficiency could be maximized.” Awkward as written, may just need to add “By” at the beginning of the sentence. Consider revising the sentence.

L29: Define the abbreviation for ULF at first use.

L 32-33: Your concluding sentence states that protein supplementation did not affect reproductive development. Adding puberty attainment or conception data to the abstract would help support this claim.

L41-42: Remove “USDA NASS, 2019” to fix citation.

L 129: Please add how long after the end of treatments heifers underwent the second synchronization protocol.

L 131-132: Please add pregnancy detection method and timing. This will aid the reader in understanding the collection of data for Table 6 and the long-term effects of protein supplementation on pregnancy rates.

L393-396: Consider revising your concluding sentence for this section to better characterize the conclusions of the result you are presenting in the specific section (similar to how you formatted your previous sections). You are presenting results evaluating the long-term effects of protein supplementation on reproduction by looking at post-supplementation pregnancy rates in this section. Detected a significant difference in the survival analysis for time to conception, with a 22-day difference between CON and P20 heifers. Your concluding sentence for this section should support that.

Table 5: Consider rewriting the title to better characterize the data presented. Clarifying that the correlation is between uterine fluid AA concentrations during development and AA concentrations in the ULF post-insemination would help the table be stand alone.

Author Response

Thank you for your time and effort in reviewing our manuscript. Please find our edits and responses to your comments.

Reviewer 2 Report

"This study highlights the importance of replacement heifer development and the influence of diet on the uterine environment in beef production. While protein supplementation did not significantly impact uterine amino acid concentrations, it was found that these concentrations changed during development. Moreover, protein supplementation affected uterine luminal fluid composition on day 14 post-insemination, potentially influencing conception rates. Further research is needed to fully understand the implications of these findings and develop effective management strategies."

Specific comments:

Title: I suggest adding the breed in the title

Simple summary: I suggest to rewrite the simple summary following the guidelines of the journal (avoid abbreviations and references)

Line 48-49 I suggest citing this recent paper: 10.1080/1828051X.2022.2032850

Lines 49-51: I suggest citing this recent paper: 10.3389/fvets.2023.1141286

I suggest avoiding to cite graduation thesis as references (8)

Material and methods

Report age, ethical approvement, ad libitum report in ithalics,

Lines 99-101: please provide more information and methods regarding feed analysis

Report feed characteristics in a table, as now it is hard to read

Statistical analysis: report the model formula to a better clarity of the method

Conclusions:

"The conclusions presented in this study lack clarity and fail to provide a strong and coherent summary of the findings. The statement regarding the impact of protein supplementation on body weight (BW) and body condition score (BCS) without affecting age at puberty is vague and lacks concrete evidence. The authors acknowledge unintended variations in final weights, which raises concerns about the reliability of the results. Furthermore, the suggested importance of the overall nutritional plane on heifer development lacks sufficient supporting evidence and fails to address potential limitations or confounding factors.

The call for further research to determine the effects of lower weights at puberty achieved through different supplementation strategies on reproductive success is reasonable, but no specific recommendations or directions for future studies are provided. The statement that supplement type affects progesterone (P4) levels and amino acid (AA) concentrations in the uterine luminal fluid (ULF) is interesting, but no clear implications or practical applications are discussed.

In conclusion, the current conclusions lack depth and fail to provide meaningful insights or actionable recommendations. The paper would greatly benefit from a more comprehensive analysis of the findings, better organization of the conclusions section, and clearer connections between the results and their implications for reproductive and production efficiencies in beef cattle.

Author Response

(The authors gave the same response as above.)

Reviewer 3 Report

The Research Article entitled: "The impacts of supplemental protein during heifer development on amino acid concentrations in the uterus and pregnancy outcomes" studied the role of different levels of protein supplementation in affecting the uterine environment of beef heifers without inhibiting development or the ability to conceive. The authors quantified the AA concentrations and pH in ULF. The experiments are well designed and the authors have performed a large number of analyses. The statistics with correlations are also complete.

I think the paper is interesting and suitable to be published in Animals Journal after minor revisions:

1.      Matherials and Methods section:

a.      In sub section “2.3 Progesterone and Puberty” line 153: “Blood samples” or plasma or serum samples?

b.      In sub section “2.3 Progesterone and Pubertywhat is the detection limit of P4 assay? Specify.

c.      In sub section “2.4 Amino Acid Analysis” what was the column used for LC/MS? and chromatographic conditions?

2.      Results section:

a.      In sub section “2.3 Progesterone and Puberty” lines 239 and 244: plasma P4????? but in materials and methods you mentioned serum. Are the collection tubes for serum …?

b.      figure 4:  blood plasma????

Review the text and correct these contradictions

Author Response

(The authors gave the same response as above.)

Round 2

Reviewer 2 Report

Dear authors,

I'm glad to hear that the revised version of your paper, titled "The impacts of supplemental protein during heifer development on amino acid concentrations in the uterus and pregnancy outcomes," has improved significantly. It's always encouraging to see authors make enhancements based on feedback.

I would like to congratulate you on addressing the issues raised during the review process and making substantial improvements to your manuscript. The inclusion of additional references, the expansion of the material and methods section, and the clarification of Table 4 have certainly enhanced the quality and clarity of your study.

I appreciate your efforts in providing a descriptive statistics section to provide a clear overview of the population included in the study. This addition will greatly benefit readers in understanding the characteristics and composition of the study sample.

Moreover, the inclusion of practical implications and limitations in the discussion section adds valuable insights to the research. It helps readers understand the real-world implications of your findings and the potential constraints or areas for further investigation.

Overall, the revisions made to your paper have strengthened its scientific rigor and improved its readability. I believe the changes you have implemented have enhanced the overall quality of the study and will contribute to the advancement of knowledge in the field.

Thank you for your hard work and dedication to improving your manuscript. I recommend submitting the revised version for publication consideration.

Best regards,